# Real-Time Intelligent Detection System for Illegal Wearing of On-Site Power Construction Worker Based on Edge-YOLO and Low-Cost Edge Devices

Rong Chang [1], Bangyuan Li [1], Junpeng Dang [1], Chuanxu Yang [1], Anning Pan [2,3,*] and Yang Yang [4]

1 Yuxi Power Supply Bureau, Yunnan Power Grid Corporation, Yuxi 653100, China
2 School of Big Data, Baoshan University, Baoshan 678000, China
3 School of Physics and Electronic Information, Yunnan Normal University, Kunming 650500, China
4 School of Information Science and Technology, Yunnan Normal University, Kunming 650500, China
* Correspondence: anning@ynnu.edu.cn

**Abstract:** Ensuring personal safety and preventing accidents are critical aspects of power construction safety supervision. However, current monitoring methods are inefficient and unreliable as most of them rely on manual monitoring and transmission, which results in slow detection and delayed warnings regarding violations. To overcome these challenges, we propose an intelligent detection system that can accurately identify instances of illegal wearing of power construction workers in real-time. Firstly, we integrated the squeeze-and-excitation (SE) module into our convolutional neural network to enhance detection accuracy. This module effectively prioritizes informative features while suppressing less relevant ones, resulting in improved overall performance. Secondly, we present an embedded real-time detection system that utilizes Jetson Xavier NX and Edge-YOLO. This system promptly detects and alerts power construction workers of instances of illegal wearing behavior. To ensure a lightweight implementation, we design appropriate detection heads based on target size and distribution, reducing model parameters while enhancing detection speed and minimizing accuracy loss. Additionally, we employed data augmentation to enhance the system's robustness. Our experimental results demonstrate that our improved Edge-YOLO model achieves high detection precision and recall rates of 0.964 and 0.966, respectively, with a frame rate of 35.36 frames per second when deployed on Jetson Xavier NX. Therefore, Edge-YOLO proves to be an ideal choice for intelligent real-time detection systems, providing superior accuracy and speed performance compared to the original YOLOv5s model and other models in the YOLO series for safety monitoring at construction sites.

**Keywords:** Edge-YOLO; YOLOv5s; edge device; power construction

## 1. Introduction

In the context of power construction, safety accidents resulting from unsafe behaviors among staff are common occurrences. In complex construction site scenarios, hidden safety hazards may arise when workers neglect to wear safety equipment, which poses a serious threat to their safety and lives [1]. Compliance with safety regulations, including the proper usage of safety helmets, safety belts, work clothes, and safety gloves, is of paramount importance in safeguarding the well-being of individuals engaged in construction work. Despite the existence of safety regulations, enforcement among workers often falls short, which poses challenges in ensuring safety at construction sites. Consequently, power grid enterprises face difficulties in monitoring and verifying workers' compliance with these regulations. Identifying and warning workers about non-compliance is crucial to maintaining safety on construction sites.

In recent years, video surveillance cameras have been widely adopted in substation construction safety management systems [2]. However, traditional detection methods that

rely on manual labor exhibit several drawbacks, such as high rates of missed detection, low efficiency, labor-intensiveness, and low reliability [3]. These drawbacks are largely attributed to the complex backgrounds and terrains of Yunnan province, where substation construction takes place. To address the challenges of slow transmission speed and lack of timely warning for violations, real-time detection of compliance wearing is necessary to prevent accidents. Deep learning-based object detection algorithms have been increasingly applied in recent years to replace manual monitoring of on-site safety regulations. These algorithms offer a more reliable and efficient way to ensure the safe operation of the power grid [4]. Furthermore, recent advances in drone and camera phone technology have made it possible to capture video sequences with moving cameras for detecting and tracking of moving objects [5]. These object detection algorithms rely on cloud platforms with substantial computing power or high-performance GPU computer clusters to achieve superior detection rates. However, GPU chips have limitations in terms of server power consumption, size, and portability, making them unsuitable for front-end scenarios. Therefore, low-cost edge devices offer a viable solution by simplifying hardware complexity and providing efficient and effective power construction safety detection in real-world settings.

At a construction site that requires significant power, video acquisition equipment is connected to the substation's intelligent management system via a high-speed 4G or 5G network. This advanced technology allows the substation's remote safety construction management and control system center to capture live video feeds and closely monitor every aspect of the construction process. However, environmental constraints such as incomplete 4G or 5G signal coverage and signal delays in certain field construction environments must be taken into consideration. For instance, Yunnan Province exhibits a complex topography, with mountains and plateaus comprising over 90% of its landscape. As a result, signal delays can easily trigger electric shock hazards and high-altitude accidents. Furthermore, traditional safety measures primarily focus on post-incident actions rather than proactive prevention. Therefore, real-time violation detection is crucial in reducing casualties. In addition, given the special environmental requirements, the model developed in this study is deployed based on edge devices. Edge devices provide the necessary localized processing for time-sensitive operations and can provide local versions of cloud services to maintain system functionality in the event of a cloud server disconnection.

Object detection is a crucial task in computer vision, and numerous methods and algorithms have been proposed to address this issue. However, most of these methods are designed for detecting and analyzing a single wearable object, which limits their effectiveness in the multi-specification wearable application environment of power systems. To improve the detection accuracy of targets in power systems and to more effectively locate illegal wearing, advanced target detection algorithms are often combined with traditional or binary classification algorithms. However, these models tend to be relatively complex and struggle to achieve a balance between speed and accuracy. Additionally, there has been a lack of emphasis on the development of real-time detectors for low-cost edge hardware, such as the Jetson Xavier NX. To address these limitations, it is urgent to propose an advanced multi-wearable object recognition and defect location method that can effectively solve the aforementioned problems. In this paper, we investigate an improved intelligent detection method based on Edge-YOLO for detecting instances of illegal wearing among on-site power construction workers. Our proposed approach embeds lightweight models into the edge devices, achieving a balance between precision and real-time network architecture.

Our main contributions are summarized as follows: (1) Our proposed Edge-YOLO model enhances object detection accuracy by integrating the SE [6] module, which adds a channel attention mechanism to the bottleneck structure of the backbone network for improved feature extraction. This module prioritizes informative features and suppresses less relevant ones, enabling it to perform well on edge devices for automatically detecting complete compliance with wearing requirements for power construction site workers; (2) We present a novel embedded real-time detection system, leveraging Jetson Xavier NX and Edge-YOLO, for the purpose of identifying and alarming instances of illegal wearing

behavior among power construction workers. To achieve a lightweight implementation of the embedded model, we adopt a strategy of designing suitable detection heads that take into consideration the size and distribution of the targets. This approach aims to reduce the model's parameter count while simultaneously improving detection speed, all while minimizing potential accuracy reduction; (3) A data augmentation method has been proposed to increase the number and diversity of the dataset. We collect original images from real power construction sites or networks and enhance the resulting dataset to be used for training our models.

The remainder of this paper is organized as follows: Section 2 introduces the related work. In Section 3, we present our improved detection methods for various objects. Section 4 provides detailed explanations of our data augmentation methods. Section 5 outlines the experiments we conducted to evaluate our method. Finally, in Section 6, we present our conclusions.

## 2. Related Work

Wearing compliance on power construction sites is an important measure to ensure the safety of power grid construction. Previously, substation supervision mainly relied on manual monitoring. However, with the development of surveillance technology, human supervision is gradually being replaced by video surveillance. Nevertheless, video monitoring tasks still require manual observation of the camera system, leading to challenges in achieving real-time and accurate feedback due to the extensive time required for monitoring multiple surveillance cameras. In recent years, intelligent video analysis has emerged as a promising approach to replace traditional methods to support electric power enterprises in behavior recognition and prediction, employee safety, perimeter intrusion detection, and vandalism deterrence. To leverage the benefits of video monitoring systems for the aforementioned purposes, a computer vision-based automatic solution is necessary to enable real-time detection of the substantial volume of unstructured image data collected.

Traditional object detection algorithms, such as ViolaJones detector (VJ.Det) [7], histogram of oriented gradient (HOG.Det) [8], and the deformable part-based mode [9], adopt window sliding and manual feature extraction methods. However, these methods have limitations in adapting to multiscale features and tend to exhibit low detection efficiency and accuracy. In recent years, deep learning-based object detection methods have been proposed and continuously applied, taking advantage of the ongoing development of deep learning. Therefore, many object detection algorithms based on deep learning have been introduced, including AlexNet [10] and residual learning [11]. Based on their approaches, current object detection methods using deep learning can be categorized into two-stage and one-stage object detection. The former includes faster R-CNN [12], FPN [13], Certernet [14], etc. While achieving high accuracy, the previously mentioned two-stage object detectors often exhibit low efficiency, making real-time detection challenging. In contrast, one-stage target detectors, such as the YOLO (You Only Look Once) series [15], RetinaNet [16], EfficientNet [17], etc., strike a balance between real-time detection and accuracy. Additionally, lightweight models are designed for mobile devices and other computing resource-constrained environments, such as drones and edge devices [18,19]. Among these models, SSDLite [20] is a typical architecture used in lightweight object detection. The MobileNet series [21] gradually improve the performance of the constructed model with depthwise separable convolutions, while Xception [22] is designed to enhance network performance without increasing network complexity.

Moreover, several breakthroughs have been achieved in the field of object detection, particularly with the YOLO family of detectors. Many researchers have succeeded in reducing the size of YOLO models, enabling real-time detection. YOLO-LITE [23] offers a more efficient model for mobile devices, while YOLObile [24] introduces block-punched pruning and mobile acceleration using a collaborative scheme between mobile GPUs and CPUs. However, research on real-time object detection for safety construction monitoring is still in its early stages. YOLOv5s is an improvement of the YOLO series for lightweight networks,

which provides a trade-off between detection speed and precision [25]. Yan et al. [26] proposed a system that combines remote substation construction management and artificial intelligence object detection techniques during construction in real-time based on YOLOv5s. Several researchers have conducted studies on related variants of YOLOv5 to improve the model's performance. For example, Liao et al. [27] built the device components using a simple online and real-time tracking and counting algorithm on pruned YOLOv5. Liu et al. [28] improved YOLOv5n by optimizing the configuration of the target detector head and the network structure, solving the problems of low efficiency and redundant parameters in feature extraction in the model. Additionally, Xu et al. [29] proposed a target detection algorithm based on the YOLOv5 algorithm to address the issues of low accuracy and strong interference in existing safety helmet wearing detection algorithms and successfully improves the detection accuracy of safety helmets. The algorithm effectively demonstrated that by adding the SE (squeeze-and-excitation) block module to the YOLOv5 model, it is not only possible to obtain the weights of image channels but also to accurately separate the foreground and background of the image. By contrast, we aim to further improve YOLOv5s models [30] for real-time detection on low-cost edge devices in the study for real-time safety monitoring of electric power enterprises. We deployed the model to low-cost edge devices, ultimately achieving accurate and fast detection of compliance wearing in power grid construction. The study focuses on the automatic detection of complete compliance wearing for power construction site workers, including safety helmets, safety belts, work clothes, and safety gloves, rather than just detection a single item.

## 3. Materials and Methods

### 3.1. Jetson Xavier NX

The Jetson Xavier NX is an edge device that can serve as either an end device or an edge server capable of performing computing tasks on the device itself [31]. It is designed for high-performance AI systems such as drones, portable medical devices, small commercial robots, automated optical detection, and other IoT-embedded systems. It is designed to deliver powerful deep learning computations while maintaining portability. In addition, the Jetson Xavier NX has a WIFI module on the back for connecting to the host system via a wireless network. This device offers several advantages, including high computational efficiency, rapid response time, low energy consumption, and cost-effectiveness, among others. In this study, the YOLOv5s-based lightweight model proposed is embedded into the Jetson Xavier NX to provide efficient and effective solutions for power construction safety detection. Table 1 shows the parameters of the Jetson Xavier NX.

**Table 1.** Parameters of Jetson Xavier NX.

| Items | Parameters |
| --- | --- |
| GPU | 384-core NVIDIA Volta $^{TM}$ with 48 Tensor cores |
| CPU | six-core NVIDIA Carmel ARMv8.2 64-bit CPU |
| Memory | 16 GB 128-bit LPDDR4x@1600MHz |
| Data storage | 16GB eMMC5.1 |
| CSI | CSI support for up to 6 cameras (14 channels) MIPI CSI-2 D-PHY 1.2 |
| PCIE | $1 \times 1$ (PCIE3.0) + $1 \times 4$ (PCIE4.0), 144 GT/s |
| Video encoding/decoding | 2 K × 4 K 60 Hz encoding (HEVC); 2K4K60Hz decoding |
| Size | 69.6 mm × 45 mm |

### 3.2. The Edge-YOLO Architecture

The objective of this study is to detect compliance violations in the safety gear of power construction workers, including safety helmets, safety belts, work clothes, and safety gloves. Given that the system is intended to be deployed on edge devices, the real-time performance of the detection model is critical. Additionally, due to the unique nature of power construction safety detection, the system must operate continuously once

activated. Thus, the accuracy and stability of the model become indispensable. In this paper, we present an innovative deep learning network architecture, Edge-YOLO, which is an improved version of YOLOv5s. Our proposed model is specifically designed for real-time monitoring of power construction sites and is capable of detecting non-compliant worker behavior related to safety regulations.

The YOLOv5 algorithm is a popular object detection model optimized for GPU computing. It is the fifth iteration of the original YOLO algorithm and includes several variations based on the width and depth of the network. The latest version, YOLOv5 7.0 [32], includes YOLOv5n, YOLOv5s, YOLOv5m, YOLOv5l, and YOLOv5x. Among these variations, the YOLOv5s model has been specifically designed to streamline the network and achieve high speed on edge devices, such as the Jetson Xavier NX, with a frame rate of up to 30 FPS. Additionally, the YOLOv5s model offers several advantages, including high detection accuracy, stability, and ease of deployment. Notably, it outperforms YOLOv5n, YOLOv7-tiny, and YOLOv8n in terms of both accuracy and stability. Therefore, a model based on YOLOv5s is well-suited for deployment on the Jetson Xavier NX and is an ideal choice for an intelligent power construction compliance wearing detection system. The YOLOv5s model consists of three primary components: backbone, neck, and prediction (as illustrated in Figure 1). The entire working process of the YOLOv5s network can be described in detail as follows.

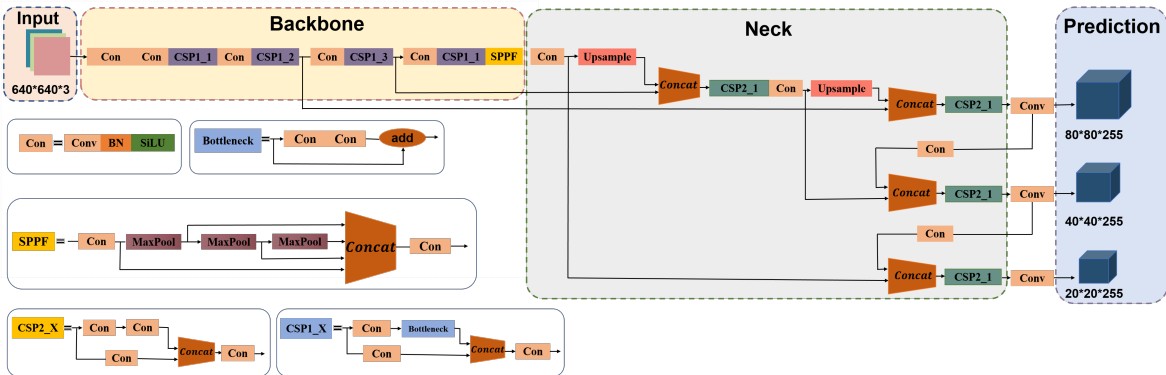

**Figure 1.** The YOLOv5s structure.

(1) Backbone: The backbone network comprises a Con structure, Batch Normalization (BN), and SiLu operations. It includes CSP1_X blocks and a SPPF (Spatial Pyramid Pooling-Fast) module. CSP1_X blocks concatenate different convolutions to extract valuable features from the input image. The network applies a Convolution module with a size of 3 and a step size of 2 to extract image features, producing an output image of size $160 \times 160 \times 128$. Three groups of CSP1 and Conv layers further enhance feature extraction, resulting in a feature map of size $20 \times 20 \times 1024$. A bottleneck structure improves detection accuracy. An SPPF module is incorporated to enhance the accuracy of the $20 \times 20$ feature map, capturing features at multiple scales and improving object detection accuracy.

(2) Neck: The Neck module, comprising CSP1_X and CSP2_X, plays a critical role in balancing network speed and accuracy. CSP2_X reduces model parameters while improving precision. Additionally, the Neck also includes an upsampling process that upsampled feature maps of size $80 \times 80 \times 512$ using two sets of CSP2_X, Conv, Upsample, and Concat connections of size 1 and step 1. As a result, the network produces feature maps of $80 \times 80 \times 512$, $40 \times 40 \times 512$, and $20 \times 20 \times 512$ after subsampling the initial $80 \times 80 \times 512$ feature map.

(3) Prediction: The prediction module of YOLOv5s adopts a multi-scale feature map approach for comprehensive object detection. Convolutional (Conv) operations generate three feature maps of different scales: $80 \times 80 \times 255$, $40 \times 40 \times 255$, and $20 \times 20 \times 255$. These feature maps enable the model to detect both small and large targets within a single image. By utilizing Conv operations and generating candidate boxes on these feature maps, the model accurately detects the presence and location of objects within the image.

To further improve accuracy and meet real-time requirements for edge device optimization, this paper proposes an improved model named Edge-YOLO based on the YOLOv5s framework. Figure 2 illustrates the overall architecture of Edge-YOLO. In the backbone structure, we have added the SE module into the bottleneck structure of YOLOv5s to improve detection accuracy. In the head structure, we have designed appropriate detection heads to reduce the model's parameter size. Specifically, we designed the largest header to significantly increase detection speed without a noticeable decline in accuracy, meeting the speed requirements of edge equipment.

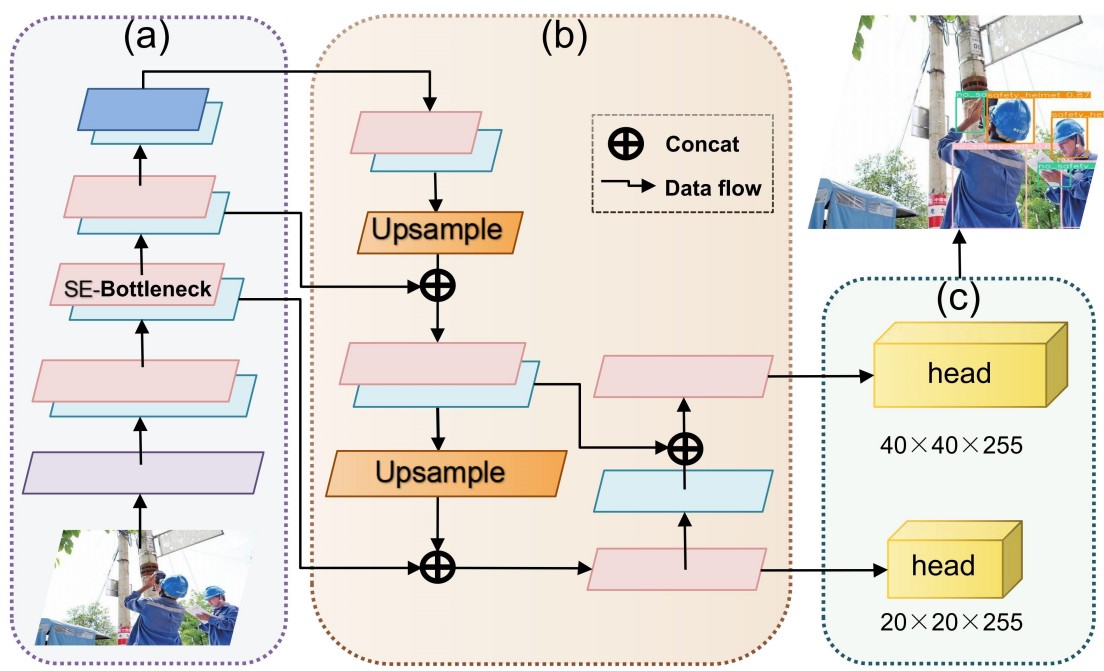

**Figure 2.** Illustration of the overall Edge-YOLO architecture. (**a**) Backbones, (**b**) Necks, (**c**) Heads, and detection input and results.

To ensure high accuracy while meeting the resource constraints of edge devices, we investigated the impact of reducing the detection head of YOLOv5s. We found that the detection accuracy decreased as the size of the head was reduced. To mitigate this decline, we introduced the Squeeze-and-Excitation (SE) module to refine the features. By incorporating the SE module, we were able to improve the detection accuracy while still maintaining a small model size that is well-suited for deployment on edge devices.

Figure 3 illustrates the integration of the Squeeze-and-Excitation (SE) module, which serves as a channel attention mechanism. The SE module is added to the original residual block as an additional pathway. It computes the initial channel weights via the global pooling layer and subsequently refines them for each channel using two fully connected layers and a sigmoid activation function. Finally, the original channel is multiplied by the weights for each channel, leading to improved detection accuracy during the learning and training process through the gradient descent method in the network. The formula for the squeezing operation is shown below:

$$g_c = F_s(u_c) = \frac{1}{H \times W} \sum_{i=1}^{H} \sum_{j=1}^{W} u_c(i,j), g \in R^c \tag{1}$$

where $F_s$ is the squeeze operation, $u$ denotes the input space, and $U \in R^{H \times W \times C}$, and $c$ represents each channel. Furthermore, the excitation formula is given as follows:

$$t = F_e(g, W) = œ(W_2 Sigmoid(W_1 g)) \tag{2}$$

where $F_e$ presents the excitation operation, $W_1 \in R^{\frac{C}{r} \times C}$, $W_2 \in R^{C \times \frac{C}{r}}$, $r$ denotes a hyperparameter with a default value of 16, indicating the dimension reduction coefficient of the first fully connected layer. Additionally, the formula of scale is shown as follows:

$$\widetilde{u} = F_{sc}(u_c, t_c) = u_c \times t_c \tag{3}$$

where $F_{sc}$ shows the scale operation. The SE module plays a crucial role in learning the weight coefficients to effectively distinguish features in the model. The embedding of the SE module can significantly enhance the detection accuracy of the model.

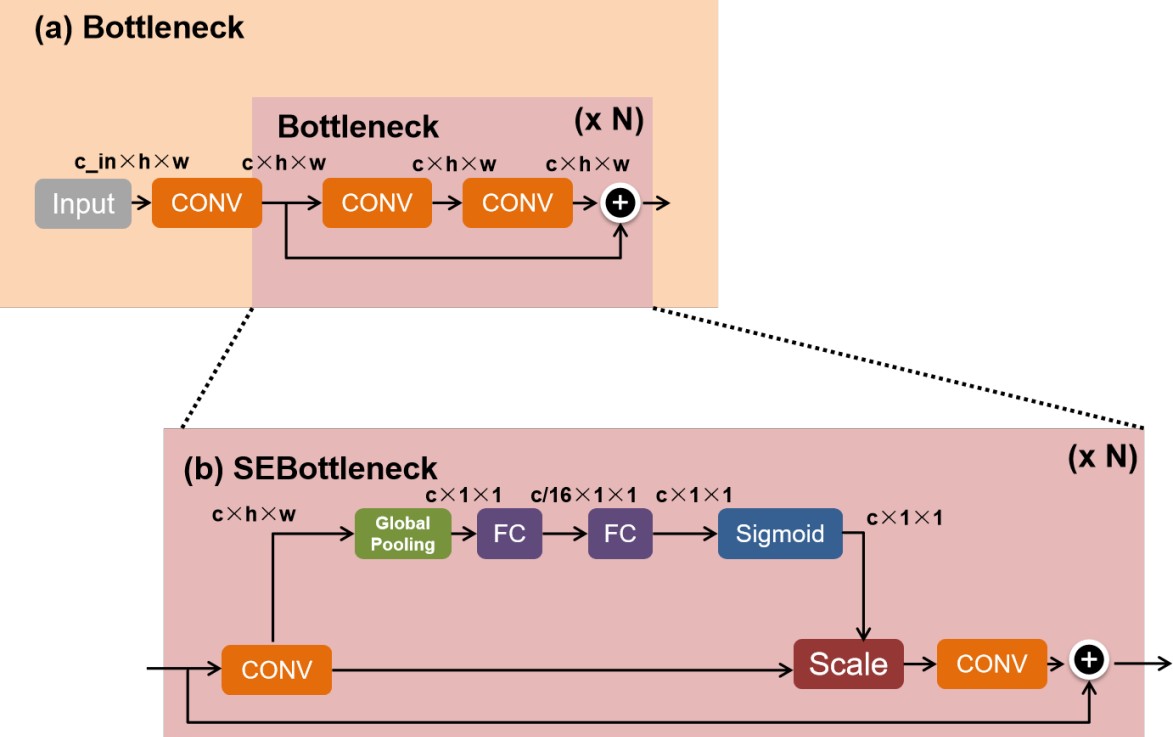

**Figure 3.** (**a**) Bottleneck structure of YOLOv5s, (**b**) Bottleneck structure embed with SE module.

### 3.3. Optimizing Model Deployment and Application on Jetson Xavier NX

In a conventional detection system, the process of capturing images or videos through a camera and transmitting the video signal to a server over the network can be time-consuming. Subsequently, the detection of illegal wearing by power construction workers relies on a detection model deployed on the server, with the detection results and alarm signals transmitted back to the client. However, this process introduces network delays and alarm delays, thereby compromising the reliability of standard wearing detection. To address these limitations, we have deployed the trained Edge-YOLO model on the powerful Jetson Xavier NX-embedded device. This embedded device has been strategically deployed in various substations. Leveraging the computational capabilities of Jetson Xavier NX, the need for server deployment and network transmission has been eliminated. Consequently, real-time detection and early warning of illegal wearing by power construction workers have become achievable. This deployment paradigm significantly improves the reliability and efficiency of standard wearing detection for power construction workers, offering a more practical solution for real-time monitoring and timely intervention in substations.

The deployment of the Jetson Xavier NX algorithm involved its seamless integration with the NVIDIA DeepStream SDK. To accomplish this, we followed a series of systematic steps as outlined below, the detection system flowchart is shown in Figure 4.

(1) Preparation of test dataset and operating environment setup: We meticulously prepared the required test dataset for evaluation purposes. In addition, based on DeepStream YOLO and the Edge-YOLO model, we established the necessary operating environment.

(2) Trained model import and file conversion: We imported a trained "pt" model into the system for deployment on Jetson Xavier NX. Through the conversion process, we obtained the configuration (cfg) and weight (wts) files required for further proceedings.

(3) Compilation of libraries and configuration file editing: We compiled the necessary libraries to ensure compatibility with the deployed algorithm. As per specific requirements, we meticulously edited the configuration files, fine-tuning them for optimal performance.

(4) Model testing and comparative analysis: We rigorously conducted tests on the deployed model to assess its performance in real-world scenarios. Through comparative analyses, we evaluated the detection results, comparing them against established benchmarks.

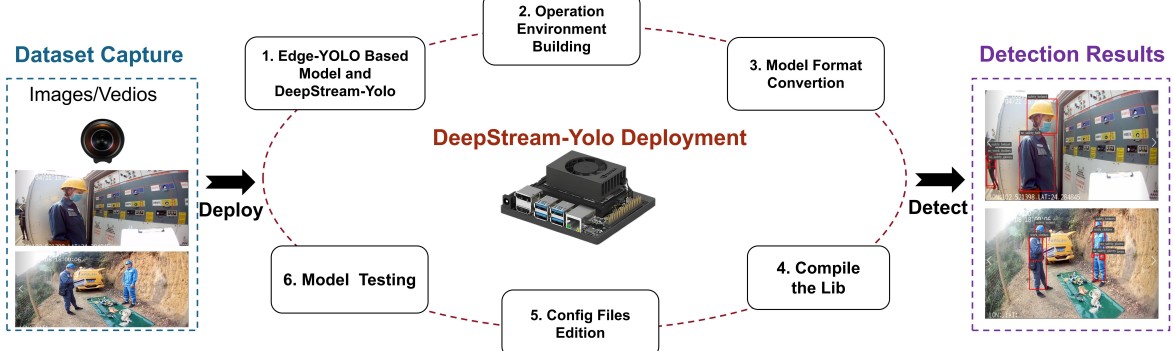

**Figure 4.** The detection system flowchart.

Upon successful deployment of the trained YOLOv5.pt model to the Jetson Xavier NX, the current embedded device can be utilized as a comprehensive intelligent detection system for identifying illegal wearing practices among power construction workers. This system can be deployed in various substations without the need for a host, enabling real-time detection of safety helmet, safety belt, work clothes, and safety gloves without relying on a network connection. The proposed method for detecting illegal wearing among power construction workers comprises the following three stages:

(1) Data Acquisition Phase: The Jetson Xavier NX-embedded device is deployed in substations to capture real-time video streams or images using a monocular camera connected to the device.

(2) Image Pre-processing Phase: Video streams and images are processed by the Jetson Xavier NX using the YOLOv5s training model. Leveraging the powerful computing capabilities of the Jetson Xavier NX, this stage ensures accurate and real-time standardized wearing detection for power construction workers without the need for host support.

(3) Detection Result Display and Non-standard Dress Warning: A high-definition display and voice warning device can be connected to the Jetson Xavier NX via the USB terminal or wireless network. This setup allows for real-time monitoring of electric workers' illegal wearing practices through video or voice alerts.

## 4. Data Augmentation

The dataset utilized in the experiment consists of four components: safety helmet, safety belt, work clothes, and safety gloves. These components encompass both positive and negative samples. During the data collection process, the diversity of sample data was taken into account. However, due to the high number of data classes, obtaining a sufficient

amount of data can be challenging, particularly for the standard wearing of Yunnan Power Grid Corporation's work clothes and safety gloves.

During the training process of our neural network model, a small dataset can lead to overfitting, which weakens the model's ability to generalize. Overfitting occurs when the model performs well on the training set but performs poorly on the test set. To avoid this issue, appropriate data augmentation methods can be used to increase the variety of images in the dataset and achieve a more reasonable data distribution. This approach can help prevent overfitting and enhance the model's generalization performance.

To account for the effects of equipment at the power construction site and environmental factors during filming, this paper employs four primary techniques for data augmentation: (1) merging object segmentation with the background; (2) affine transformations; (3) definition transformations [26]; (4) brightness and contrast transformations. Each operation simulates a range of real-life surveillance video scenarios encountered during on-site power construction, including changes in viewing angles, variations in distance, fluctuations in backgrounds, differences in image clarity, and adjustments in illumination.

### 4.1. Merging Object Segmentation with Background

The training dataset can be divided into two categories based on their role during training: foreground information and background information. The foreground information includes the detection targets such as safety helmets, safety belts, work clothes, and safety gloves, while the background information refers to other image information. Due to the lack of real-world power construction scenes, the available dataset are limited. To meet the volume requirements of the training dataset and enhance the performance of the trained model, we utilized a method of fusing foreign objects and background to augment the construction dataset. Specifically, we collected background images taken in realistic power construction sites without any workers present, and then performed segmentation on images from various scenarios to extract the workers under construction. These segmented workers were subsequently fused into the power construction environment [26]. Figure 5 presents an example of using the existing dataset to generate a new dataset through segmentation and fusion.

### 4.2. Affine Transformation

Affine transformations were applied to modify the size, orientation, and position of safety helmets, safety belts, work clothes, and safety gloves in the original images and fused images, simulating the conditions of power construction sites. Geometric transformations such as rotation, scaling, translation, and zooming were used to augment the dataset, generating additional image samples. The robustness of detection models can be improved by utilizing generated images for training. For instance, a power construction scene captured by a video camera can be simulated from different angles, resulting in a diverse set of training data. By incorporating these generated images into the training process, the detection model can learn to recognize and adapt to a wider range of scenarios, ultimately leading to improved robustness.

### 4.3. Definition Transformation

We utilize Gaussian noise as a data augmentation technique to simulate sensor noise resulting from poor lighting and/or high temperature during data acquisition. The primary objective is to introduce noise to elements within the image of the source dataset, resulting in an image with added noise. To accomplish this, we utilize a Gaussian kernel function to perform a convolution on the image, transforming it into a blurred image. The integration of artificially generated images during the training phase can lead to a considerable enhancement in both accuracy and stability of detection models.

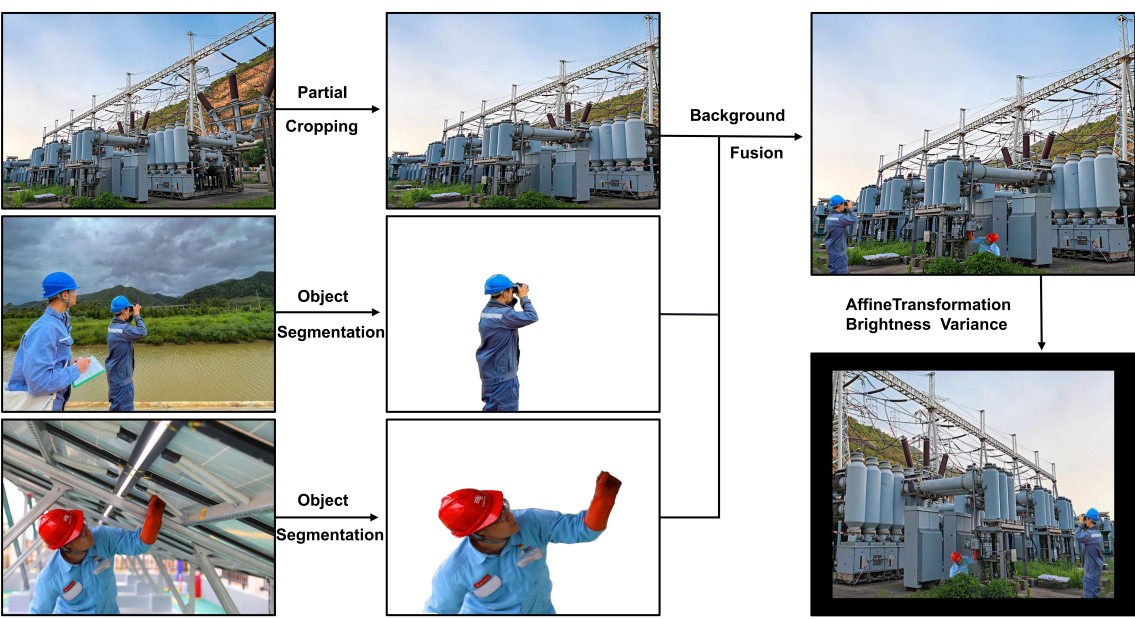

**Figure 5.** Segmentation and fusion process.

### 4.4. Brightness and Contrast Transformation

In the field of deep learning research, brightness and contrast transformations are commonly categorized into linear and nonlinear transformations. These transformations are applied to generate diverse dataset that can simulate real-world changes in illumination resulting from different weather conditions. This approach aims to enhance the robustness of detection models when dealing with varying illumination conditions.

## 5. Results and Discussion

### 5.1. Dataset and Experiment Environment

In this study, we employ two methods for data collection, namely web crawling and photography of the construction site of Yunnan Power Grid Corporation workers in Yuxi City, Yunnan Province. To enhance the detection rate, confidence, and robustness of the model, a data augmentation method is utilized to optimize the dataset. The parameter settings for data augmentation are shown in Table 2. After data enhancement, the dataset comprises a total of 10,702 images, which are categorized into eight classes, namely (1) wearing a safety belt; (2) no safety belt wearing; (3) wearing a helmet; (4) no helmet wearing; (5) wearing work clothes; (6) no work clothes wearing; (7) wearing safety gloves; (8) no safety gloves wearing. The dataset has been split into a training dataset and a validation set with a ratio of 8:2, resulting in 8562 images for the training dataset and 2140 images for the validation dataset. In addition, we have also collected an additional 400 images of the power construction scene as the test set.

To assess the real-time and effectiveness of intelligent compliance detection methods, we integrated Edge-YOLO deep learning models into Jetson Xavier NX. Additionally, a series of experiments have been conducted on the deep learning framework Pytorch to evaluate the detection performance of various inspection methods. To train our model, we utilize a computer equipped with a NVIDIA RTX-3080 GPU boasting 24 GB of memory, an Intel Core i7-10700 CPU, and 32 GB of RAM to handle the intensive computational requirements of our research. In addition, our system is deployed on the Ubuntu operating system.

**Table 2.** Parameters setting of data augmentation.

| Parameter Items | Value |
|---|---|
| rotation scale | 16° |
| zoom range | 0.7–1.2 |
| horizontal flip range | 0.6 |
| brightness transformation range | 0.6–1.5 |
| Gaussian noise probability | 0.4 |
| Gaussian Kernel size | 3 |

### 5.2. Evaluation Criteria

We evaluated the performance of different detection models using five commonly used evaluation metrics: Precision ($P$), Recall ($R$), $F_1$ Score ($F_1$), mean Average Precision ($mAP$), the number of frames transmitted per second (Frames Per Second, FPS), and the number of floating-point operations (FLOPs). The formulae for $P$, $R$ , $F_1$, and $mAP$ are defined as follows:

$$P = \frac{T_P}{T_P + F_P} \tag{4}$$

$$P = \frac{T_P}{T_P + F_N} \tag{5}$$

$$F_1 = \frac{2 * P * R}{P + R} \tag{6}$$

$$AP_i = \int_0^1 P(R)dR \tag{7}$$

$$mAP = \frac{1}{N} \sum_i^N AP_i \tag{8}$$

where $F_P$ (false positive) and $F_N$ (false negative) represent the number of incorrect and missed detected objects, respectively, while $T_P$ (true positive) indicates the number of correctly detected objects. $T_P + F_P$ represents the total number of objects detected, while $T_P + F_N$ represents the total number of actual objects. $F_1$ score indicates the harmonic mean of $P$ and $R$. $AP$ refers to the area under the precision-recall curve (PR curve), and $mAP$ is the average of APs for different classes, and $N$ is the number of classes in the test sample. In object detection tasks, a higher precision value indicates fewer false detection results, while a higher recall value denotes fewer missed detection results.

### 5.3. Implementation Details

Our objective is to achieve precise and real-time identification of on-site electrical construction workers' wear using an enhanced model, Edge-YOLO. The Edge-YOLO model is specifically designed for real-time processing and aims to enhance portability and reduce dependency on infrastructure. To accomplish this, we plan to deploy the model on a Jetson Xavier NX edge device, which offers high-performance computing capabilities with low power consumption. To evaluate detection speed and accuracy, we trained six additional variants of YOLO models: YOLOv5n, YOLOv5s, YOLOv5m (three variants of YOLOv5 with different network sizes), YOLOv7, YOLOv7-tiny, and YOLOv7x (three models of the YOLO algorithm). We trained our models using YOLO pre-training weights and utilized the dataset we constructed for both training and validation. We ensured that the training parameters for each model remained consistent throughout the training process. Table 3 provides detailed training parameter settings for the YOLO models used in this series.

**Table 3.** Training parameters setting of detection method.

| Parameter Items | Value |
|---|---|
| Epoch | 300 |
| Batch size | 8 |
| Worker | 8 |
| Momentum | 0.95 |
| Initial learning rate | 0.001 |

*5.4. Effect of Model Optimization*

We conducted various strategies to improve the real-time performance and accuracy of our intelligent detection system. Firstly, we performed a series of optimization operations on the dataset. Building upon that, we further optimized the training process and experiment results for the YOLOv5s model. Subsequently, we embedded the trained model into Jetson Xavier NX with the expectation of achieving promising results in real-world power construction scenarios. As the experiment involved detecting eight target categories, the model optimization stage focused on optimizing the dataset trained on the YOLO model. We employed image augmentation techniques to enhance performance. We performed data augmentation in this experiment to modify certain features of the images using techniques such as translation, zoom, rotation, and ten other augmentation operations. To optimize the dataset effectively, we randomly selected three operations to enhance the dataset during each image transformation process. The training accuracy of the model was evaluated before and after data augmentation, and the results are presented in Table 4. As shown in the table, the precision, recall, and mAP values were improved after data augmentation. These enhancements further enhanced the detection accuracy of the model in power-related scenarios. Moreover, to enhance the detection accuracy of the YOLOv5s model, we made advancements by integrating the SE module into the bottleneck structure. This integration, denoted as "YOLOv5s+SE", has proven effective in improving the overall performance of the model. To optimize the speed and light weight of our model, we made selective sacrifices to the precision performance of the "YOLOv5s+SE" model. Specifically, during the connection phase between the Neck and Prediction components, we meticulously devised suitable detection heads that take into account the size and distribution of the detection targets. This Edge-YOLO aims to enhance detection speed while minimizing any potential decrease in accuracy. Notably, we eliminated the largest header in the head structure to further accelerate detection and meet the requirements of edge devices. The YOLOv5s+SE model originally possessed 7,032,725 parameters, resulting in a training model speed of 2.7 ms. However, by incorporating SE and optimizing the detection heads, we managed to reduce the parameter count of Edge-YOLO to 5,241,454 and achieve a smaller training model size of 2.2 ms. The details are shown in Table 5. In addition, the table also showed the precision, recall, mAP, FPS, parameters, and the number of floating-point operations (FLOPs). Based on the performance comparison and analysis of the aforementioned models, Edge-YOLO achieves a balanced compromise between the "YOLOv5s+SE" and "YOLOv5s-header" models in terms of the number of FLOPs and parameters.

**Table 4.** Training results of the model before and after data augmentation.

| Items | P | R | mAP@.5 | mAP@.5:.95 |
|---|---|---|---|---|
| Source | 0.897 | 0.904 | 0.931 | 0.583 |
| Augmentation | 0.958 | 0.959 | 0.980 | 0.709 |

**Table 5.** Comparative experiments on improving model accuracy and speed.

| Models | P | R | $F_1$ | mAP@.5 | mAP@.5:.95 | Speed (ms) | FPS | FLOPs |
|---|---|---|---|---|---|---|---|---|
| YOLOv5s | 0.958 | 0.959 | 0.958 | 0.980 | 0.709 | 2.5 | 32.50 | 14.6 G |
| YOLOv5s+SE | 0.969 | 0.972 | 0.968 | 0.989 | 0.729 | 2.7 | 30.09 | 17.2 G |
| YOLOv5s-Header | 0.954 | 0.955 | 0.954 | 0.980 | 0.690 | 1.9 | 36.66 | 13.1 G |
| Edge-YOLO | 0.964 | 0.966 | 0.965 | 0.983 | 0.718 | 2.2 | 35.36 | 14.2 G |

### 5.5. Performance of Different Detection Models

After optimizing the entire dataset, the same validation dataset was used to compare the detection performance of seven models in terms of both speed and accuracy. The objective of this paper was to find a YOLO model that is more suitable for embedding in edge devices and better aligned with the requirements of actual power construction site inspection detection. The training precision trend of the seven models during the first 200 epochs is presented in Figure 6, where Edge-YOLO maintained relatively stable accuracy across 200 rounds of training and demonstrated an improvement in accuracy when compared to the YOLOv5s model. Furthermore, Table 6 demonstrates that the YOLOv5n and YOLOv7-tiny models have a simpler structure and faster detection speed compared to the other models. According to the performance comparison and analysis of the models in the table, the number of FLOPs and parameters of YOLOv5n and YOLOv7-tiny are less than YOLOv5s, which will save some memory space. However, their detection precision and recall are poor, with YOLOv7-tiny in particular showing the worst accuracy. The YOLOv7x, YOLOv7, and YOLOv5m models, on the other hand, have more complex structures with a large number of FLOPs and parameters, demonstrating superior detection accuracy. Nevertheless, their detection speed is slower, making them unsuitable for embedding in edge devices that require real-time performance and greater portability. Compared to both YOLOv7 and YOLOv5 series models, the YOLOv5s model offers faster detection speeds without compromising on Precision, Recall, mAP@.5, and mAP@.5:.95 metrics. Figure 7 shows the detection results of these models in the test dataset, indicating that the YOLOv5s model performs comparably to other more complex YOLO series models, but with significantly better detection accuracy than the lighter YOLOv5n and YOLOv7-tiny models. After carefully evaluating the accuracy performance close to YOLOv5m and the speed performance close to YOLOv7-tiny on edge devices, we have chosen the YOLOv5s model as our detection model to enhance compliance with real-time detection systems for on-site power construction workers. To achieve this, we proposed a novel object detection model called Edge-YOLO, which is built upon the YOLOv5s object detection algorithm and offers improvements in both detection speed and accuracy. In Figure 7, we additionally showcase the detection performance of the safety gloves to illustrate the aforementioned results. The utilization of red boxes in the figure serves to differentiate between the various types of detectors. Based on the results of safety-gloves detections, it can be observed that the YOLOv5n and YOLOv7-tiny models exhibit lower accuracy, increased false positives, and missed detections. In comparison, the detection performance of Edge-YOLO is superior to them.

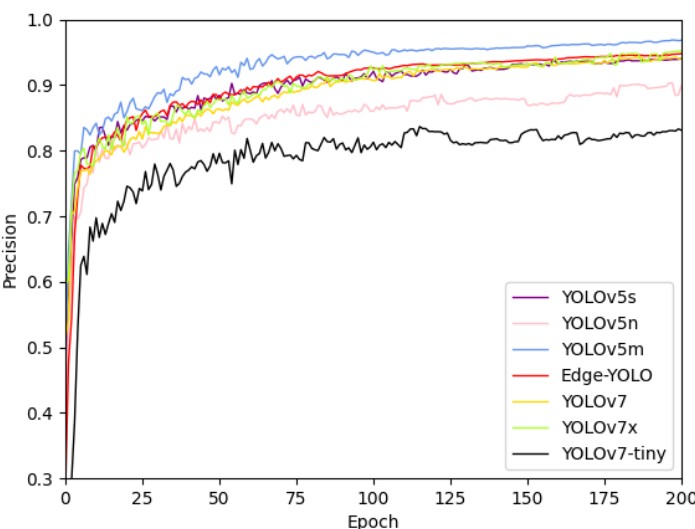

**Figure 6.** Contrast of the training precision trend.

**Table 6.** Performance of different detection models.

| Models | P | R | mAP@.5 | mAP@.5:.95 | Speed (ms) | Parameter (M) | FLOPs |
|---|---|---|---|---|---|---|---|
| YOLOv5s | 0.958 | 0.959 | 0.980 | 0.709 | 2.5 | 13.7 | 14.6 G |
| YOLOv5m | 0.978 | 0.984 | 0.991 | 0.812 | 4.2 | 40.2 | 45.3 G |
| YOLOv5n | 0.911 | 0.897 | 0.937 | 0.587 | 1.6 | 3.64 | 4.5 G |
| YOLOv7 | 0.968 | 0.972 | 0.987 | 0.765 | 7.9 | 71.3 | 96.6 G |
| YOLOv7X | 0.973 | 0.980 | 0.991 | 0.813 | 12.8 | 135.0 | 202.7 G |
| YOLOv7-tiny | 0.869 | 0.860 | 0.903 | 0.564 | 2.1 | 11.7 | 12.1 G |
| Edge-YOLO | 0.964 | 0.966 | 0.983 | 0.718 | 2.2 | 12.0 | 14.2 G |

*5.6. Results in Realistic Power Construction Sites*

In this section, we integrated seven YOLO models into the edge device Jetson Xavier NX and evaluated their processing rates in various real-world scenarios to compare the detection performance of each model. The frame-per-second (FPS) results are presented in Table 7. Our experiment indicates that the Edge-YOLO model achieved an FPS of 35.36 on the Jetson Xavier NX, demonstrating the real-time performance of our system.

**Table 7.** FPS of different detection models.

| Models | FPS |
|---|---|
| YOLOv5s | 32.50 |
| YOLOv5m | 14.20 |
| YOLOv5n | 59.90 |
| YOLOv7 | 7.40 |
| YOLOv7X | 4.30 |
| YOLOv7-tiny | 39.20 |
| Edge-YOLO | 35.36 |

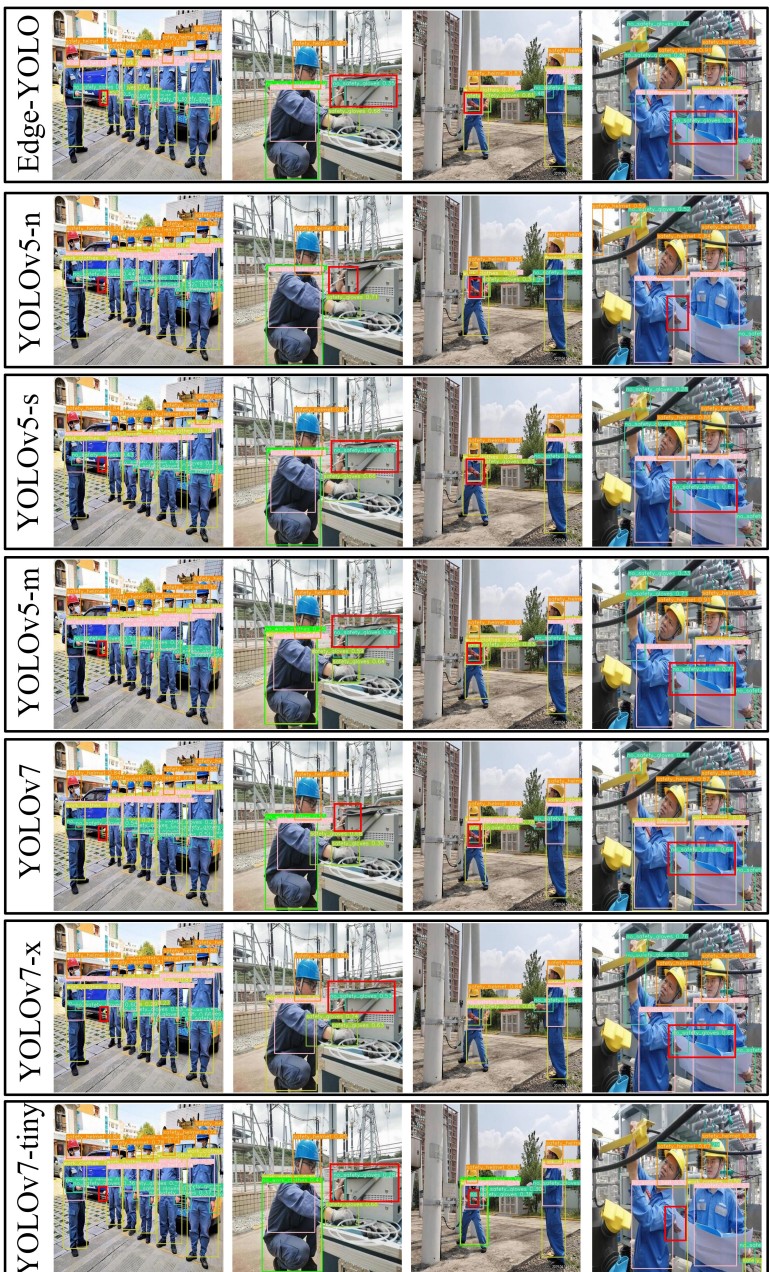

**Figure 7.** The detection performance of different models. The orange bounding box denotes the operators with a safety helmet, the light orange bounding box denotes the operators without a safety helmet, the red bounding box denotes the operators with a safety belt, the pink bounding box denotes the operators without a safety belt, the brown bounding box denotes the operators with work clothes, the light green bounding box denotes the operators without work clothes, the green bounding box denotes the operators with safety gloves, the cyan bounding box denotes the operators without safety gloves. The red boxes are employed to differentiate among the various types of detectors.

Based on the performance comparison of various models, it can be observed that the Edge-YOLO model achieves faster detection speeds than the YOLOv5s model while maintaining satisfactory accuracy. Therefore, by embedding the Edge-YOLO model on the portable Jetson Xavier NX, we can detect the compliance of workers' protective gear more effectively in real power construction sites. To assess the robustness of our system, we conducted inspections at three power construction sites with different scenarios, demonstrating the effectiveness of our intelligent edge system. The on-site inspection dataset

includes a diverse set of inspection objects. The three scenarios were: (1) high altitude operations, (2) night operations, and (3) snowy climate operations. Figure 8 shows the deployment platform of our edge device system, while Table 8 and Figure 9 present the detection results of the test dataset. Our results demonstrate the successful detection of all eight classes in various realistic power construction scenarios, showcasing excellent detection performance across different categories and operation scenarios. It can be seen that the Edge-YOLO-based model embedded into the Jetson Xavier NX for on-site detection of compliant wearable devices among power construction workers can maintain good detection accuracy. Moreover, it offers real-time performance, making it more convenient and practical for real-world scenarios.

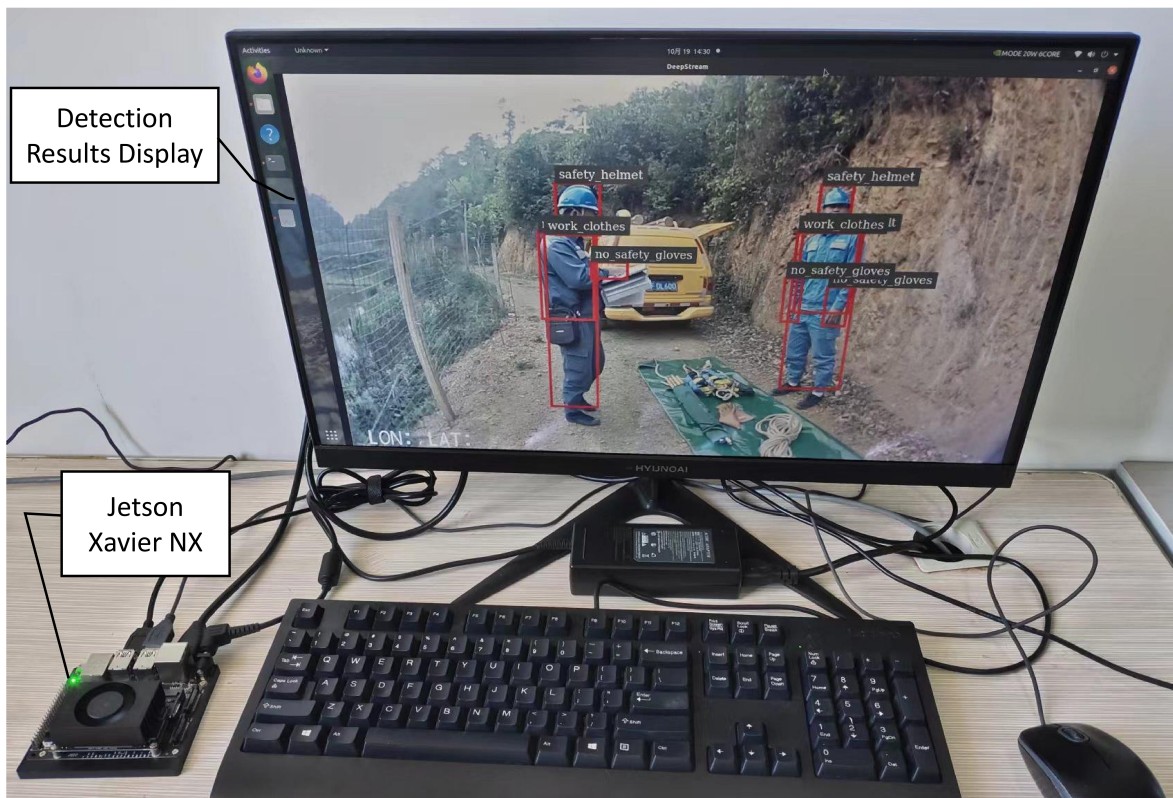

**Figure 8.** The deployment platform based on Edge-YOLO model and Jetson Xavier NX.

**Table 8.** Performance of Edge-YOLO on the actual dataset.

| Classes | P | R | mAP@.5 | $F_1$ |
|---|---|---|---|---|
| safety_belt | 0.958 | 0.923 | 0.964 | 0.940 |
| no_safety_belt | 0.971 | 0.982 | 0.990 | 0.976 |
| safety_helmet | 0.993 | 0.995 | 0.995 | 0.994 |
| no_safety_helmet | 0.978 | 0.993 | 0.994 | 0.985 |
| work_clothes | 0.950 | 0.959 | 0.981 | 0.954 |
| no_ work_clothes | 0.943 | 0.942 | 0.978 | 0.942 |
| safety_gloves | 0.962 | 0.974 | 0.985 | 0.968 |
| no_safety_gloves | 0.953 | 0.961 | 0.978 | 0.957 |
| all | 0.964 | 0.966 | 0.983 | 0.965 |



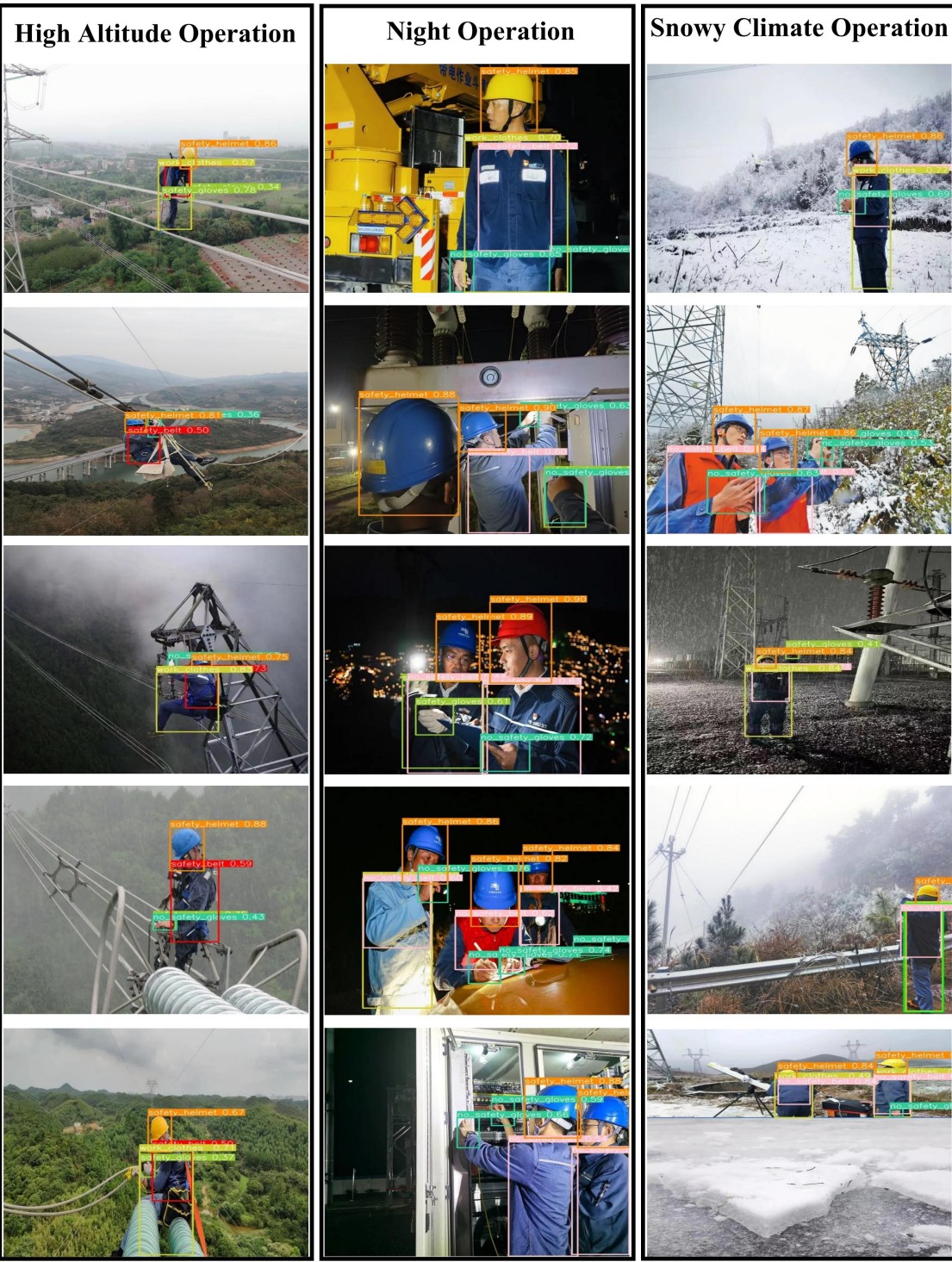

**Figure 9.** The detection performance of different scenarios. The orange bounding box denotes the operators with a safety helmet, the light orange bounding box denotes the operators without a safety helmet, the red bounding box denotes the operators with a safety belt, the pink bounding box denotes the operators without a safety belt, the brown bounding box denotes the operators with work clothes, the light green bounding box denotes the operators without work clothes, the green bounding box denotes the operators with safety gloves, the cyan bounding box denotes the operators without safety gloves.

## 6. Conclusions

The paper presents an intelligent real-time detection system for detecting wearing violations among on-site power construction workers. The system utilizes the Edge-YOLO model and incorporates it into low-cost edge devices, specifically the Jetson Xavier NX. To improve the accuracy and reduce the model size, an SE attention mechanism module has been integrated into the network. Additionally, removing the largest header from the YOLOv5s model significantly reduces the number of trainable parameters. To address the issue of insufficient training samples, a data augmentation method has been proposed to enhance object detection performance in practical power construction scenarios. The experimental results demonstrate that the proposed system can detect illegal wearing behavior with real-time performance and accuracy, as evidenced by testing on three different types of power construction site dataset. Furthermore, the deep learning-based wearing violation detection method presented in this paper has been successfully deployed and is currently operating stably.

**Author Contributions:** Conceptualization, R.C. and A.P.; methodology, R.C. and A.P.; software, B.L. and Y.Y.; validation, J.D. and C.Y.; formal analysis, R.C. and A.P.; investigation, J.D. and C.Y.; resources, C.Y. and Y.Y.; data curation, B.L. and C.Y.; writing—original draft preparation, R.C. and A.P.; writing—review and editing, R.C. and A.P.; visualization, J.D. and A.P.; supervision, R.C. and Y.Y.; project administration, R.C. and B.L. All authors have read and agreed to the published version of the manuscript.

**Funding:** The science and technology program "Research on lightweight of intelligent device and software in edge computing" funded by China Southern Power Grid provided funding for this effort. This work was also partially supported by Yunnan Province Ten thousand Talents Program.

**Data Availability Statement:** The data presented in this study are conditionally available upon request to the corresponding author.

**Acknowledgments:** The authors give thanks to the experimental platform provided by the Laboratory of Pattern Recognition and Artificial Intelligence, and also thanks to the dataset for power construction safety detection provided by Yuxi Power Supply Bureau, Yunnan Power Grid Corporation.

**Conflicts of Interest:** The authors declare no conflict of interest.

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
