# Peer review of "Real-Time Intelligent Detection System for Illegal Wearing of On-Site Power Construction Worker Based on Edge-YOLO and Low-Cost Edge Devices"

_applsci, doi:10.3390/app13148287_

Round 1

Reviewer 1 Report

1. The paper's topic is to use the algorithm for real-time detection and low-cost edge devices, so the abstract and contributions should reflect the problem of how to reduce the cost of the algorithm and how to apply it to the device.

2. The introduction of YOLOV5 is too much, thus neglecting the innovation of the article;

3. The article should focus on how to apply the algorithm to mobile devices, and the paper obviously focuses on SE attention network and image augmentation, which are not innovative.

4.  The SE module is also added to Edge-yolo, why the speed is improved instead, please explain in Table 5;

5. It is suggested to add Flops to enhance the persuasive power of the algorithm at a low cost;

6.  It is suggested that clear marks be used in Figure 7 to distinguish the difference between different types of detectors.

It is suggested to check and modify the grammar of the article

Reviewer 2 Report

The authors have put forward a real-time system aimed at identifying instances where workers, particularly those at power construction sites, are missing some of their Personal Protective Equipment (PPE). They propose the use of a deep learning network architecture called Edge-YOLO, which is an enhanced version of YOLOv5s.

It is worth noting that a different paper by Z.P. Xu et al. in 2022, titled "Safety Helmet Wearing Detection Based on YOLOv5 of Attention Mechanism," published in the Journal of Physics: Conference Series, Volume 2213, Issue 1, Pages 012038 (DOI: 10.1088/1742-6596/2213/1/012038 integrates the SE module to achieve the same objective, and I believe it should be cited somewhere along with an explanation of the differences between their approach and the proposed one.

Regarding the Introduction:

At line 22, the citation [1] is not suitable for substantiating the claim that "fail to wear safety equipment, which poses a serious threat to their safety and lives." Could you please verify the citations? Is citation [2] intended to be exchanged with [1]?

At line 31, the authors state, "However, the traditional detection methods that rely on manual labor suffer from several drawbacks, such as high rates of missed detection, low efficiency, labor-intensiveness, and low reliability," and they cite [3], which does not support this statement. On the contrary, the authors of [3] argue, "The experimental results have demonstrated that the proposed hybrid model (CNN + LSTM) can automatically extract and classify unsafe behaviors (i.e., those associated with climbing a ladder) using conventional video with a high level of accuracy." Please review the citations in the Introduction.

Figure 3 quality can be improved.

Section 6 should be titled "Conclusions" only, as the previous section contains a discussion.

Overall considerations:

The authors' proposed solution is promising because it demonstrates the ability to detect multiple objects at a multiscale level, specifically for multiple workers present in the scene. The noteworthy aspect of achieving these results on a low-cost Edge device for machine learning inference is highly appreciated. Although the inclusion of the SE module in the neural network architecture and the utilization of a Data Augmentation strategy are not groundbreaking innovations on their own, their application to the authors' specific objective, accompanied by illustrated results, makes the work compelling for publication.

Round 2

Reviewer 1 Report

After the revised paper, I think it is acceptable, no other comments.

We suggest minor revisions to inappropriate English grammar.